# Genome-Wide Identification of Bilberry WRKY Transcription Factors: Go Wild and Duplicate

**DOI:** 10.3390/plants12183176

**Published:** 2023-09-05

**Authors:** Winder Felipez, Jennifer Villavicencio, Valeria Oliveira Nizolli, Camila Pegoraro, Luciano da Maia, Antonio Costa de Oliveira

**Affiliations:** 1Instituto de Agroecología y Seguridad Alimentaria, Facultad de Ciências Agrárias, Universidad San Francisco Xavier de Chuquisaca—USFX, Casilla, Correo Central, Sucre 1046, Bolivia; winder.felipezz@gmail.com; 2Plant Genomics and Breeding Center, Departamento de Fitotecnia, Faculdade de Agronomia Eliseu Maciel, Universidade Federal de Pelotas—UFPel, Pelotas CEP 96010-900, RS, Brazil; jennifer.villavicencio371@gmail.com (J.V.); val.nizolli@gmail.com (V.O.N.); lucianoc.maia@gmail.com (L.d.M.); 3Carrera de Ingeniería Agroforestal, Facultad de Ciencias Ambientales, Universidad Cientifica del Sur—UCSUR, Antigua Panamericana Sur km 19 Villa el Salvador, Lima CP 150142, Peru

**Keywords:** *Vaccinium myrtillus* L., phylogeny, expression regulation, differential expression

## Abstract

WRKY transcription factor genes compose an important family of transcriptional regulators that are present in several plant species. According to previous studies, these genes can also perform important roles in bilberry (*Vaccinium myrtillus* L.) metabolism, making it essential to deepen our understanding of fruit ripening regulation and anthocyanin biosynthesis. In this context, the detailed characterization of these proteins will provide a comprehensive view of the functional features of *VmWRKY* genes in different plant organs and in response to different intensities of light. In this study, the investigation of the complete genome of the bilberry identified 76 *VmWRKY* genes that were evaluated and distributed in all twelve chromosomes. The proteins encoded by these genes were classified into four groups (I, II, III, and IV) based on their conserved domains and zinc finger domain types. Fifteen pairs of *VmWRKY* genes in segmental duplication and four pairs in tandem duplication were detected. A cis element analysis showed that all promoters of the *VmWRKY* genes contain at least one potential cis stress-response element. Differential expression analysis of RNA-seq data revealed that *VmWRKY* genes from bilberry show preferential or specific expression in samples. These findings provide an overview of the functional characterization of these proteins in bilberry.

## 1. Introduction

The bilberry (*Vaccinium myrtillus* L.), also known as the European wild blueberry, is a fruit that grows in the wild from the forests of Northern Europe to the Caucasus and towards the northern Asia-Pacific [1,2,3]. *V. myrtillus* is one of the 400 species belonging to the genus *Vaccinium* of the Ericaceae family, which also includes cultivated blueberry species and wild fruits such as *V. corymbosum*, *V. angustifolium*, *V. virgatum*, and *V. macrocarpon* [4]. Blueberries are usually diploid (2n = 2x = 24), but polyploid subspecies can be found in North America [2]. Due to their high levels of beneficial nutrients and bioactive phytochemicals present in the skin, pulp, and leaves, especially anthocyanins, the demand for blueberries has been increasing in the market [3,5]. Anthocyanins are responsible for their colors of red, blue, purple, and black fruits [5,6]. In the last two decades, several studies have linked the polyphenols and high natural anthocyanin levels of blueberries to the prevention of chronic diseases such as cancer, coronary heart disease, and neurological disorders [5].

In addition, there has been an increased interest in the *Vaccinium* genus, and as a result, genomic and transcriptomic resources for this genus have been developed. However, despite this progress, there is still incomplete genome assembly and annotation data, as is the case of blueberries (*V. corymbosum*) [7]. On the other hand, efforts to understand the genomic regions that control the anthocyanin composition of the berry have led to the generation of the complete genome, as is the case with bilberry (*V. myrtillus*) [4]. Anthocyanins and flavonoid compounds present in blueberry fruits have been associated with structural genes and transcription factors (TFs) in many species [2]. Advances in genomic resources and the understanding of *Vaccinium* spp. have made it possible to describe the MYBA TF loci and identify the main regulatory genes of this family that determine anthocyanin production [4]. Recent studies have shown that anthocyanin profiles in *Vaccinium* spp. are regulated by prevailing light and temperature conditions [8,9]. Blue light induces anthocyanin accumulation in pear fruit (*Pyrus communis x pyrifolia* cv Red Zaosu) and activates anthocyanin-production-responsive genes [10]. In blueberry, the light intensity regulation of anthocyanin accumulation represents a valuable data set to guide future functional and crop improvement studies [11]. In bilberry, it was demonstrated that blue and red/blue lights have the ability to promote anthocyanin biosynthesis by inducing the expression of key structural genes and accumulation of metabolites involved in the anthocyanin synthesis pathway, as well as the relationship between photosynthesis under different light qualities in blueberry leaves [12]. Several regulatory genes, including WRKY family TFs, have been proposed to control fruit development and ripening processes, as well as their relationship with responses to various stresses [13,14].

The WRKY TFs are one of the largest families of transcriptional regulators in plants and are present in several plant species [15,16]. These TFs are characterized by the WRKY domain, which contains about 60 amino acids that may be located near the N- or C-terminal regions and is composed of a highly conserved WRKYGQK heptapeptide DNA-binding sequence, followed by a zinc finger motif that binds to specific cis-regulatory elements called W-boxes (TTGACT/C) [17,18,19,20,21]. Evidence suggests that W-boxes are present in the promoter region of genes related to plant innate immunity [20]. Based on the number of WRKY domains and the pattern of zinc finger motifs, *WRKY* genes can be divided into four groups (I, II, III, and IV) [22,23,24]. Group I WRKYs have two domains containing a C2H2 zinc finger motif. Group II representatives have only one WRKY domain and one C2H2 zinc finger motif and can be divided into five subgroups (IIa, IIb, IIc, IId, and IIe). Group III WRKYs also have a single domain, but their zinc finger motif is C2HC [22,23] Recently, Group IV was identified, composed of incomplete WRKYs that do not fit into the previous groups because they lack the zinc finger motif [23,24].

We can find several examples of *WRKY* genes that act in the biosynthesis of anthocyanins, plant pigments that are present in leaves, flowers, fruits, and roots. Ultraviolet-B (UV-B) radiation promotes the synthesis of anthocyanins in many plants, and several transcription factors respond to UV-B radiation [25]. In *M. domestica*, MdWRKY72 TF increases anthocyanin synthesis by direct and indirect mechanisms when induced by UV-B radiation [26]. The overexpression of MdWRKY75 led to an increase in anthocyanin levels by binding to the MYB transcription factor promoter, MdMYB1 [27]. Other examples are the MscWRKY12 and MscWRKY19 TFs from *M. sylvestris*, which are involved in leaf pigmentation during the development of this organ [24], and the involvement of WRKY TFs in the synthesis of anthocyanins in *Raphanus sativus* L., the carmine radish [28]. Especially in fruits, such as *Pyrus* spp., their red skin accumulates anthocyanin, and the genes involved can contribute to the improvement of their appearance, as is the case with the combination of PyWRKY26 and PybHLH3 capable of co-directing the *PyMYB114* promoter to generate anthocyanin accumulation in red pears [29]. Eight PcWRKYs also participate in color development in red fruits of *P. communis*, and color fading in some fruits is due to reduced biosynthesis, increased degradation, and the suppression of anthocyanin transport [30]. In addition, by having a better understanding of the genes involved in this biosynthesis, it is possible to overexpress them. In *M. domestica*, the overexpression of MdWRKY11 [31] and MdWRKY71-L [32] plays a role in the synthesis of anthocyanins in red fruits and demonstrates that TFs participate in the UV-B signaling pathway to regulate the accumulation of anthocyanin in apple. Just like in the apple, the red coloration of mango (*Mangifera indica* L.) peels results from anthocyanin accumulation, where it was found that MiWRKY1 and MiWRKY81 were upregulated during the light induction accumulation of anthocyanin in mango, indicating that these genes may regulate anthocyanin biosynthesis [33].

The growing interest in the nutraceutical properties of blueberries makes it essential to deepen our understanding of fruit ripening regulation and anthocyanin biosynthesis. In this context, WRKY transcription factors have been identified as a promising tool for manipulating stress tolerance and the synthesis of nutritionally desirable compounds. Therefore, it is crucial to identify the WRKY proteins, from their structural classification to their phylogenetic relationships, as well as their gene structure, conservation of domains and motifs, *cis* elements, chromosomal mapping, tandem and segmental duplication, and genetic divergence in the complete bilberry genome (*Vaccinium myrtillus* L.). The detailed characterization of these proteins will provide a comprehensive view of the evolution and modification of the WRKY protein family in the crop and will help to determine the functional features of *VmWRKY* genes in different plant organs and in response to different intensities of light.

## 2. Results

### 2.1. Identification of the WRKY Protein Family in Bilberry

The statistical alignment Hidden Markov Model (HMMsearch) in the bilberry (*V. myrtillus*) proteome generated 76 WRKY proteins with complementary domains. Two Q-type WRKY proteins and 69 *WRKY* genes were identified (Table 1), and seven WRKY proteins (VmWRKY14, VmWRKY28, VmWRKY29, VmWRKY32, VmWRKY44, VmWRKY48, and VmWRKY49) were considered recent duplicates due to having more than 98% similarity (Appendix A). The heptapeptide region of the WRKY domain (WRKYGQK) showed variations (WRKYGEK, WRKYG, WRKYGRK, WRKYGKK, and WRKYLQK) within the primary nomenclature of the WRKY family (PF03106). The peptide length ranged from 111 to 740 amino acids (VmWRKY42 and VmWRKY61). The coding sequences (CDS) varied from 336 to 2223 nucleotides. The predicted protein molecular weight ranged from 13,001.347 to 82,707.564 kDa. The isoelectric point varied from 4.68 (VmWRKY27) to 9.89 (VmWRKY28). The instability index ranged from 32.01 (VmWRKY55) to 76.58 (VmWRKY27), and the average hydrophobicity varied from −0.44 (VmWRKY61) to −1.212 (VmWRKY27).

### 2.2. Analysis of Cis Elements in VmWRKY Gene Promoters

The analyses of promoter sequences of 69 *VmWRKY* genes generated 108 types of putative cis-regulatory elements, which were categorized into four known and one unknown cis-regulatory action elements (Appendix A). The largest category among the known elements comprises light-responsive elements (25%), predominantly represented by cis-action G-Box, Box 4, and GT1-motif. Following that, hormone-responsive elements (9%) with cis-action ABRE, CGTCA-motif, and TGACG-motif were identified. Elements related to development (9%) were found with cis-action CAT-box, O2-site, and GCN4_motif. Furthermore, environment-responsive elements (8%) exhibited cis-action AREM, LTR, and MBS. The elements associated with promoters and binding sites (7%) were predominantly represented by cis-action CAAT-box, TATA-box, and W-box. Notably, the WRKY transcription factors are bound to the W-box to initiate transcription. Additionally, the analysis revealed several other unknown functional elements (40%) with cis-action MYB, MYC, and STRE. 

### 2.3. Phylogeny, Gene Structure, and Motif Analysis of WRKY Protein in Bilberry

The unrooted phylogenetic tree of 69 VmWRKY protein sequences (Figure 1A) displays their relationships, forming eight subgroups (I, IIa, IIb, IIc, IId, IIe, III, and IV). The largest subgroups are group I (14 members) and group IIc (14), followed by group III (12), group IIe (10), group IIb (9), group IId (4), group IIa (4), and group IV (2). In group IV, the protein sequences contain the conserved WRKY domain but lack the zinc finger motif. The primary structure of the 69 VmWRKY gene sequences (Figure 1B) shows variation in the number of introns, ranging from 0 to 9. By conducting the analysis of conserved functional motifs in the protein sequences, a total of 20 functional motifs were identified, distributed across each VmWRKY subgroup (Figure 1C). Each subgroup has specific motif patterns (Appendix A).

### 2.4. Chromosomal Localization, VmWRKY Gene Duplication, and Divergence

A total of 64 *VmWRKY* genes were mapped onto twelve chromosomes of the bilberry genome (Figure 2), and the genes *VmWRKY65*, *VmWRKY66*, *VmWRKY67*, *VmWRKY68*, and *VmWRKY69* are sequenced at the scaffold level, and their location on the chromosome is unknown. The highest number of *VmWRKY* genes was found on chromosome 12 (11 genes), followed by chromosome 8 (9 genes), chromosome 4 (8 genes), chromosomes 1 and 6 (7 genes), chromosome 3 (5 genes), chromosomes 11 and 9 (4 genes), chromosome 2 (3 genes), and chromosomes 10, 5, and 7 (2 genes). Tandem and segmental duplications were identified (Table 2). Four pairs of *VmWRKY* genes were identified as tandem duplications (<100 Kb and >70% similarity) (Appendix A), and 12 pairs of *VmWRKY* genes were identified as segmental duplications (>70% similarity) (Appendix A). This information provides valuable insights into the genetic evolution of the gene family and its domains regarding other model species such as *Arabidopsis thaliana* (Appendix A). The estimation of divergence time for paralogous *VmWRKY* gene pairs, based on their synonymous substitution rates, ranged from 0.13 to 83.40 million years for the *VmWRKY40/VmWRKY41* and *VmWRKY61/VmWRKY63* gene pairs. Other duplicated gene pairs did not yield divergence time estimates with the methods applied using DNA sequences, coding sequences (CDS), and the transcriptome annotation of the genome. Another analysis was applied, and the Nei and Gojobori (NG) method was utilized to determine the divergence between proteins (Appendix A).

### 2.5. Number of Transcripts and Expression Patterns of VmWRKY Genes in Various Tissues

The expression patterns of each transcriptomic sample under light and organ conditions were identified in fragments per kilobase million (FPKM) out of a total of 36,405 (100%) annotated genes in the bilberry reference genome (Appendix A). Gene expression in the light samples was predominantly observed in the red light condition (24,491 = 67%), followed by the control samples (23,741 = 65%) and blue light samples (23,825 = 65%). The multi-dimensional scaling (MDS) analysis of the samples revealed dissimilarity in expression patterns (Figure 3(A1)). The expression of *VmWRKY* genes was identified in nearly equal proportions in the control sample (61 = 0.17%), red light sample (61 = 0.17%), and blue light sample (59 = 0.16%). However, 56 *VmWRKY* genes were found to be common across all samples (Figure 3(A2)), while 10 *VmWRKY* genes (*VmWRKY11, VmWRKY42, VmWRKY3*, *VmWRKY43*, *VmWRKY2*, *VmWRKY19*, *VmWRKY40*, *VmWRKY33, VmWRKY26*, and *VmWRKY67*) exhibited differential expression in each sample (Figure 3(A3)). Additionally, five genes (*VmWRKY11*, *VmWRKY5*, *VmWRKY19*, *VmWRKY12,* and *VmWRKY34*) showed high expression levels across all three samples (Figure 3(A4)).

The gene expression in plant organ samples showed a higher proportion in whole berries (24,268 = 67%) and berry peels (24,449 = 67%), followed by berry pulp (24,190 = 66%), roots (23,370 = 64%), and leaves (22,791 = 63%) (Appendix A). The multi-dimensional scaling (MDS) analysis of the samples revealed similarity patterns among three samples and dissimilarity among two samples (Figure 3(B1)). The expressed *VmWRKY* genes were predominantly identified in roots (62 = 0.17%) and berry peels (60 = 0.16%), followed by leaves (57 = 0.16%), whole berries (57 = 0.16%), and berry pulp (56 = 0.15%). However, 48 *VmWRKY* genes were found to be common across all samples (Figure 3(B2)). Among the root sample, 41 *VmWRKY* genes exhibited high differential expression compared to the other samples (Figure 3(B3)). Furthermore, 15 *VmWRKY* genes were highly expressed in all samples (Figure 3(B4)). Additionally, complementary analyses of raw and processed transcriptomic sample-specific data are presented (Appendix A).

## 3. Discussion

### 3.1. Identification of the WRKY Protein Family in Bilberry

Transcription factors exist in the form of a superfamily of genes and play a critical regulatory role in plants’ growth, development, and response to the environment [34]. In recent years, there has been a significant increase in interest in investigating the WRKY protein family in different plant species, including the bilberry crop. With the publication of the complete genome of several species, the identification and analysis of TF families at the whole genome level has become one of the focuses of genomic research. The WRKY TF family plays an important role in plant growth and development and defense mechanisms [35].

In the bilberry genome, it was possible to identify 76 *VmWRKY* genes. This number is higher than those of the model species *Arabidopsis thaliana* (72) [22], *Vitis vinifera* (59) [36], and *Rubus occidentalis* (60) [37]. In contrast, in other species such as *Oryza sativa* (109) [19], *Triticum aestivum* (160) [18], *Malus domestica* cv Gala (112) [24], and *Pyrus Bretschneideri* (103) [38], a larger number of genes were identified [19]. The higher number of WRKY genes in some species may be due to polyploidy and genome size [18].

### 3.2. Analysis of Cis Elements in VmWRKY Gene Promoters

A particular feature of the WRKY TF family is the ability to specifically bind to W-box ((C/T)TGAC(T/C)). Nevertheless, they bind also to other cis-acting elements located in the promoter region of their target genes [19,36,39]. In this study, it was possible to observe that the promoter region of *VmWRKY* genes contains a total of 108 conserved cis-regulatory elements with diverse functions. Strikingly, 40 *VmWRKYs* presented one or more W-boxes, indicating the putative autoregulation of these TFs, a number that is higher than the 26 *FaWRKYs* observed in strawberry [23,40,41,42]. Common promoter and enhancer regions, such as A-Box and CAAT-box, were also identified, as well as TATA-box, which are regions located around −30 of transcription initiation. In addition, the presence of HD-Zip 1 and HD-Zip 3, the elements involved in mesophyll cell differentiation and the site where protein binding occurs, were detected.

The presence of a large number of phytohormone-responsive elements, such as TGA (responsive to auxin), TATC-box (response to gibberellin), SARE (salicylic acid-responsive element), TCA (response to salicylic acid), ABRE (response to abscisic acid), AuxRR-core (responsive to auxin), CGTCA motif (response to MeJA), TGACG motif (response to MeJA), GARE (responsive to gibberellin), and P-box (gibberellin-responsive element), was detected. Additionally, 27 promoter regions associated with the response to light were identified, with G-Box, Box 4, GT1-motif, and TCT-motif being the most abundant within the species. This coincides with what was reported in raspberry (*Rubus occidentalis*), where the most abundant promoter region was Gbox, ATC, and TCT-motif [37]. 

The *VmWRKY* gene promoter regions also showed conserved cis-regulatory elements, which are involved in a variety of functions, such as responses to biotic and abiotic stresses. Among these elements are TC-rich repeats, which act in defense and response to stress; LTRs, which respond to low temperatures; AREs, which act in anaerobic induction; GC motifs, which act in specific anoxic induction; MBSs, which act in response to water stress; the WUN motif, which responds to tissue damage; and AT-rich sequences and AT-rich elements. The presence of diverse cis-acting elements, which mediate responses to environmental stresses and plant hormones, suggests that these WRKY TFs are involved in various biological processes.

### 3.3. Phylogeny, Gene Structure, and Motif Analysis of WRKY Proteins in Bilberry

Plants have adaptation mechanisms to adverse environments that involve signal transduction and molecular regulation in response to stress. TFs play a key role in this process by activating or inhibiting gene transcription by their specific binding to gene promoter regions. This results in the induction of functional gene expression and contributes to signal transduction in response to stress [43,44].

In the phylogenetic analysis of TFs, the 69 *VmWRKY* proteins were distributed into seven clusters. Cluster 1 presents nine WRKY proteins, cluster 2 presents four, cluster 3 presents ten, cluster 4 presents four, cluster 5 presents fourteen, cluster 6 presents sixteen, and cluster 7 presents twelve. The proteins belonging to each group (I, II, III, and IV), with group II subdivided into five distinct subgroups (IIa–e), formed clusters with members of only one group, and others were made up of members of two or more groups. During the intragroup evolutionary analysis of *VmWRKY* genes, it was observed that genes from group IIc were more divergent from the other members within group II (Iia, Iib, Iid, and IIe). Within this branch, IIa and IIb were grouped separately in a branch within group II, similarly to what was found in plum [45] and black raspberry [37]. Also, IId and IIe formed another subgroup. Group I presented almost all members of this group clustered together, with two additional members from group III and group IV. The last group formed was composed of almost all members from group III but included one genotype from group IV (Figure 1). All WRKY TFs from group I in bilberry contain two WRKY domains. Among them, 15 *VmWRKY*s (21.7%) were assigned to group I, and 47 *VmWRKY* (68.12%) genes were distributed in group II, which was further classified into five subgroups, IIa, IIb, IIc, IId, and IIe, which contained 7, 9, 14, 5, and 12 *VmWRKY*s, respectively. The remaining 12 *VmWRKY*s (17.4%) belonged to group III, and 2 *VmWRKY*s (2.9%) belonged to group IV. These results are similar to those found in grape (*Vitis vinifera*) and blackberry (*Rubus occidentalis*), in which the number of *WRKY*s genes found in group II were the most abundant with 39 *VvWRKY*s (66.1%) [36] and 25 *RoWRKY*s (41.6%), respectively [37].

Group III of WRKY gene members presents a zinc finger motif different from groups I and II and can be considered the most dynamic in terms of evolution [34,46]. In this study, 12 *VmWRKYs* were identified as group III, which is similar to the 13 *AtWRKYs* found in *Arabidopsis* [47] and the 10 found in wild strawberry [48]. However, this count is higher than the 6 *VvWRKYs* found in grapes [36] and lower than the 28 *OsWRKYs* in rice [34]. Some WRKYs genes from group III are part of the signaling pathway of the plant’s defense system, having a significant impact on resistance to diseases and drought [34].

Finally, we have *VmWRKY68* and *VmWRKY69*, which were grouped in group IV for not fitting into any of the other groups because the WRKY domain has a partial or lacks a complete zinc finger motif structure. WRKY genes belonging to group IV were also reported in species such as *Arabidopsis thaliana* with two *AtWRKY* genes [22] and *Pennisetum glaucum* with twelve *PgWRKY* genes [41]. Likewise, *M. domestica* (10% = 13 *MdWRKY*s) and *Vitis vinifera* (2% = 2 *VvWRKY*s) [49] presented WRKY proteins without a complete domain. This occurrence may mean the loss of the WRKY domain [21,50,51] and modifying the functional properties of the proteins [42]. In studies performed in *Hylocereus undatus*, the absence of the WRKY domain, the zinc finger or coiled-coil sequence, did not allow the binding between the WRKY protein and the promoter region of the gene [52], suggesting that genes belonging to group IV can originate non-functional proteins.

### 3.4. Chromosomal Localization, VmWRKY Gene Duplication, and Divergence

Overall, 69 *VmWRKY* genes were identified in the bilberry genome; however, only 64 *VmWRKY* genes (*VmWRKY1*–*VmWRKY 64*) had a known location on the chromosomes, and they were distributed into twelve chromosomes. The *WRKY* genes (*VmWRKY65*−*VmWRKY69*) are sequenced at the scaffold level and have no known chromosome locations. And most of the *VmWRKY*s were abundant on Chr 12. The number of *VmWRKY* genes is similar to that of the pitaya (*Hylocereus undatus*), with 70 *HuWRKY* genes distributed on eleven chromosomes [52]. The raspberry also presents a similar number of *WRKY* genes (60 *RoWRKY*); however, it has a lower number of chromosomes (7), and Chr 6 had the largest number of *RoWRKY* genes, representing 23.33% of the *WRKY* genes [37]. In the case of grapes (*Vitis vinifera*), the number of *WRKY* genes present in this species is also similar (59 *VvWRKYs*); however, they are mapped to nineteen chromosomes [38], a greater number of chromosomes than that present in bilberry. The differential distribution of the *WRKY* genes present in a species may imply chromosomal rearrangements and duplication events that took place during its evolution [41].

It has been observed that gene duplication, including tandem duplication, fragment duplication, and genome duplication, is a key factor in gene family amplification in plant genomes [53]. In this particular study, 15 segmental duplication events and four tandem duplication events (*VmWRKY37/VmWRKY38*, *VmWRKY40/VmWRKY41*, *VmWRKY46/VmWRKY47*, and *VmWRKY48/VmWRKY49*) were identified (Table 2). A high frequency of segmental duplication was also observed in *Rubus occidentalis*, with five genes containing homologous segments [37]. In *Fragaria vesca*, segmental duplications were higher than tandem duplications, representing 84.2% and 15.8% of the total duplications, respectively. Compared to what has been observed in *V. vinifera* [54] and *Oryza rufipogon* [55], tandem duplication events can significantly contribute to the expansion of *VmWRKY* genes [52].

### 3.5. Expression Patterns of WRKY Genes by Induction of Light and Plant Organs

The processing of transcriptomic samples from bilberry plant organs induced by light showed variation in the total gene expression and specific expression of *VmWRKY* genes in each sample (Appendix A). In the case of samples treated with red light (24,491 = 67%), gene expression was higher than in the control and blue light samples. This pattern was also observed in the specific expression of *VmWRKY* genes. However, 17 *VmWRKY* genes showed differential expression under red light compared to the control and blue light (Figure 3(A3)). On the other hand, in pear (*P. communis x pyrifolia* cv ‘Red Zaosu’), exposure to blue light led to an increase in anthocyanin accumulation and the activation of genes responsible for anthocyanin production [10]. In red mango fruit (*M. indica* L.), UV-B light induction positively regulates anthocyanin accumulation, and the genes *MiWRKY1* and *MiWRKY81* are involved in this regulation [33]. In general, anthocyanin accumulation is influenced by light availability, and the specific impact of different light qualities varies among plant species [56,57,58]. These results suggest that VmWRKYs are concentrated in the skin and berry of the fruit, which may indicate an association with higher levels of anthocyanins, the main source of organoleptic and antioxidant properties in blueberries, as reported in a gene expression study during bilberry development [59]. However, the expression level in bilberry leaves was high compared to that observed in Tetrastigma (*Tetrastigma hemsleyanum*), a grape family plant with eight WRKYs, which showed high expression levels in leaves [60].

Gene expression in plant organ samples showed similar levels in the entire berry (24,268 = 67%) and the berry skin (24,449 = 67%), which were comparable to those observed in the berry pulp (24,190 = 66%) and higher than those numbers found in leaves (22,791 = 63%) and roots (23,370 = 64%) (Appendix A). However, a greater number of *VmWRKY* genes were expressed in the root sample (62 = 0.17%), and the differential expression of 41 *VmWRKY* genes was high compared to the rest of the samples (Figure 3(B3)). The higher number of genes and their differential expression in the root sample could be explained by the role played by WRKY transcription factors, which are mainly involved in development and stress responses, such as salt and water stress tolerance [61]. Since bilberries are sensitive to these conditions, it is important to pay close attention to the presence of VmWRKYs in directly affected organs such as roots, where *VmWRKY54* and *VmWRKY11* showed higher expression levels (Figure 3(B4)). This is supported by examples of the effectiveness of these transcription factors in Arabidopsis, where the overexpression of *AtWRKY46*, *GmWRKY13*, or *VvWRKY11* can positively regulate salt and water stress tolerance [62,63,64,65], and in *Nicotiana benthamiana*, where the overexpression of GhWRKY41 conferred tolerance to water and salt stress [66].

The complete genome of bilberry (*Vaccinium myrtillus* L.) allowed the characterization of 69 members of the *VmWRKY* gene family. Segmental and tandem duplications were detected and could enhance biotic/abiotic resistance in the bilberry genome. The average ages of duplications were identified as 8.27 mya (range 013–16.41) for the tandem and 26.43 mya (range 0.48–83.40) for the segmental duplications, suggesting more recent events for tandem duplications than segmental duplications.

## 4. Materials and Methods

### 4.1. Identification of WRKY Proteins in the Bilberry

The complete genome sequence of the bilberry (*Vaccinium myrtillus* isolate NK2018 v1.0 genome sequence) was downloaded from the Genome Database for Vaccinium (GDV) https://www.vaccinium.org/crop/bilberry (accessed on 17 February 2023) [4]. To identify possible candidate amino acid sequences of *VmWRKY* bilberry. The WRKY domain HMM model (PF03106) was downloaded from Pfam (http://pfam.xfam.org/family/PF03106/hmm) (accessed on 17 February 2023) [67]. The Hmmer software [68] was also used for similarity search in annotated proteins in bilberry using 1 × 10^−3^ as the upper limit of the e-value. All obtained protein sequences were examined for the presence of the WRKY domain using the Web CD Search Tool (https://www.ncbi.nlm.nih.gov/Structure/cdd/wrpsb.cgi) (accessed on 17 February 2023) [69]. The ExPASY ProtParam (https://web.expasy.org/protparam/) (accessed on 25 February 2023) [70] was used to predict the isoelectric point (pI), molecular weight (MW), and overall average hydrophobicity (GRAVY) of each *VmWRKY*.

### 4.2. Analysis of Cis Elements in VmWRKY Gene Promoters

For each *VmWRKY* gene, a 2000 bp sequence upstream of the start codon was retrieved from the bilberry (*Vaccinium myrtillus* isolate NK2018 v1.0 genome sequence) by applying integrative genomics viewer—IGV [71]. This sequence was submitted to the PlantCARE website to investigate cis-acting regulatory elements (http://bioinformatics.psb.ugent.be/webtools/plantcare/html/) (accessed on 17 February 2023) [72].

### 4.3. Multiple Alignment and Phylogenetic Analyses

A phylogenetic tree was constructed to compare bilberry WRKY proteins. Multiple alignment of WRKY protein sequences was performed with ClustalW software using standard parameters [73]. The phylogenetic tree was constructed using BEAST v.2.5 software [74], with the UPGMA clustering method [75]. A bootstrap analysis was conducted using 10,000,000 replicates, and evolutionary distances were calculated using the JTT matrix-based method [76].

### 4.4. Analysis of Gene Structure and Identification of Conservation of Motifs

To investigate the diversity and structure of *VmWRKY* family members, genomic sequences for their exon/intron were used and plotted on TBtools [77], based on bilberry genome annotation information (*Vaccinium myrtillus* isolate NK2018 v1.0 genome sequence). VmWRKY protein sequences were used to identify conserved motifs using the Expectation Maximization Tool for Motive Elicitation MEME (https://meme-suite.org/meme/tools/meme) (accessed on 20 February 2023) [78]. The parameters were as follows: number of repetitions: any; maximum number of motifs: 20; optimal motif widths: 8 to 50 amino acid residues. 

### 4.5. Chromosomal Localization, Gene Duplication, Ka/Ks Calculation, and Divergence Time Estimation

The chromosomal location image of the *VmWRKY* genes was generated by the TBtools software [77], according to chromosomal position information provided in the genomic database for Vaccinium-GDV (https://www.vaccinium.org/) (accessed on 30 February 2023). To identify specific tandem duplications of *VmWRKY*s, the following criteria were used: genes within a 100 kbp region on an individual chromosome with a sequence similarity of ≥70% [79]. The pairwise local alignment calculation of two protein sequences was performed by the Smith–Waterman algorithm of EMBOSS Water (http://www.ebi.ac.uk/Tools/psa/) (accessed on 17 February 2023) [80].

For the calculation of non-synonymous substitutions per site (*Ka*) and the number of synonymous substitutions per synonymous site (*Ks*), in addition to comparing the selection pressure, a *Ka*/*Ks* ratio greater than 1, less than 1, and equal to 1 represent positive, negative, and neutral selection, respectively. For each pair of genes, the value of *Ks* was used to estimate the time of divergence in millions of years based on a rate of 6.1 × 10^−9^ replacements per site per year, and the time of divergence (T) was calculated as T = *Ks*/(2 × 6.1 × 10^−9^) × 10^−6^ million years ago (*Mya*) [81]. The bioinformatics tool used for genetic divergence was the simple Ka/Ks Calculator (NG) from TBtools-II [77].

### 4.6. Transcriptomic Analysis of VmWRKY Genes in Bilberry 

The expression patterns of *VmWRKYs* genes were analyzed based on published RNA-seq data on NCBI BioProject ID PRJNA739815 [4]. For the analysis of differential expression, five samples of RNA-seq data were collected that were sequenced by the Illumina HiSeq4000: leaf (SRR14876435), root (SRR14876436), whole berry (SRR14876437), berry flesh (SRR14876438), and berry skin (SRR14876439). Also analyzed were RNA-seq samples from Bioproject ID PRJNA747684 [82]. The experimental design of red and blue light was as follows: control (SRR15179770, SRR15179771, and SRR15179772); 6 days of continuous irradiation (red light) (SRR15179767, SRR15179768, and SRR15179769); 6 days of continuous irradiation (blue light) (SRR15179773, SRR15179774, and SRR15179775).

Data processing was performed in the following three steps. (a) Quality control and adjustment of samples: SRA toolkit [83] was used to download the data samples, FastQC [84] was employed to analyze and visualize the quality of readings, and Trimmomatic ver. 0.39 [85] was applied to remove the low quality and library adaptors. (b) The reads were mapped against the bilberry (*V. myrtillus*) reference genome [4] using the software HISAT2 [86]. (c) For the counting of total reads aligned by gene in the different libraries, FeatureCounts [87] was used. The quantification analysis (d) was performed using the packages limma, edgeR, and DESeq2 in R software [88]. In this protocol, a FPKM normalization method (fragments per kilo base per million mapped reads) was used, which are counts scaled by the total number of reads and the expression of *VmWRKY* genes per library. The analysis of gene expression proportion was based on the number of genes annotated in the bilberry genome (36,405 genes = 100%), and the expression of each transcriptome sample and *VmWRKY* genes was proportional to the total annotated genome genes. A heatmap was produced using TBTools with normalized data (log2 counts per FPKM) [77]. Then, a multi-dimensional scaling (MDS) plot was generated to check the repeatability of the sample and the overall difference between samples. Additionally, a MeanVar plot and Biological Coefficient of Variation (BCV) were calculated.

## 5. Conclusions

The complete genome of bilberry (*Vaccinium myrtillus* L.) contains 76 identified and 69 characterized members of the *VmWRKY* gene family, which are located on twelve chromosomes. Additionally, 12 segmental and four tandem duplications were identified in the bilberry genome. The cis-regulatory elements found in the promoters of *VmWRKY* genes are putatively involved in various functions related to biotic and abiotic stress responses. The differential expression of *VmWRKY* genes was induced by light, suggesting its involvement in anthocyanin biosynthesis. The characterization of *VmWRKY* genes in the bilberry genome, their phylogenetic relationships, and differential expression are important for understanding their role in regulating anthocyanin biosynthesis and their adaptation to different environmental conditions.

## Figures and Tables

**Figure 1 plants-12-03176-f001:**
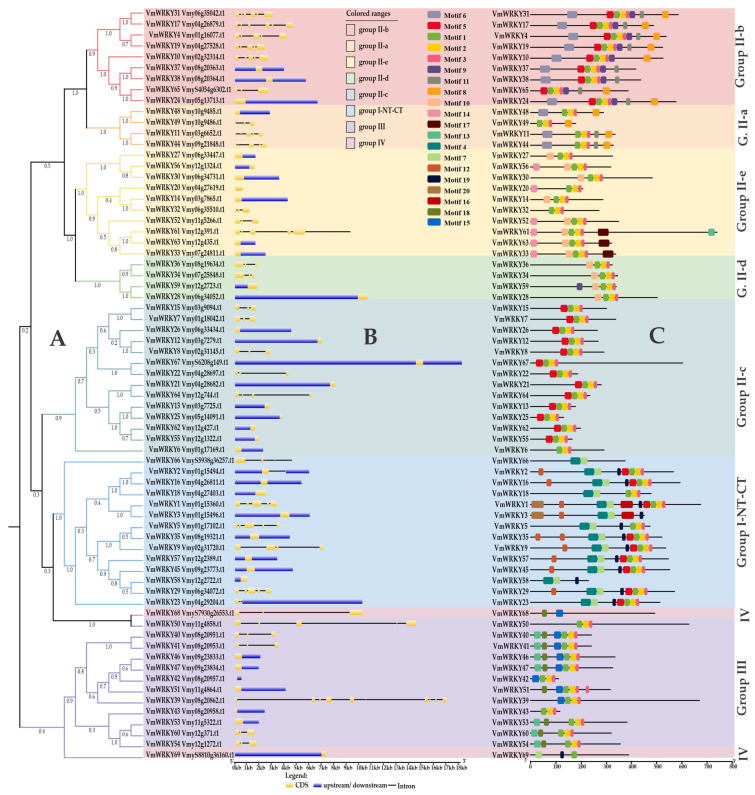
Phylogenetic relationships and structure of genes encoding the bilberry VmWRKY proteins: (**A**) The unrooted tree was generated with the BEAST program using the full-length amino acid sequences of the 69 bilberry VmWRKY proteins by the UPGMA method, with 1,000,000 bootstrap replications. VmWRKY protein subfamilies (I, IIa, IIb, IIc, IId, IIe, III, and IV). **(B**) Exon/intron organization of bilberry *VmWRKY* genes. Yellow boxes represent exons, and black lines represent introns. Untranslated regions (UTRs) are indicated by blue boxes. The sizes of exons and introns can be estimated using the scale at the bottom. (**C**) Schematic representation of conserved motifs in bilberry VmWRKY proteins, elucidated from publicly available data (NCBI CDD Domain–Pfam–18,271 PSSMs). Each colored rectangular box represents a motif with the given name and motif consensus.

**Figure 2 plants-12-03176-f002:**
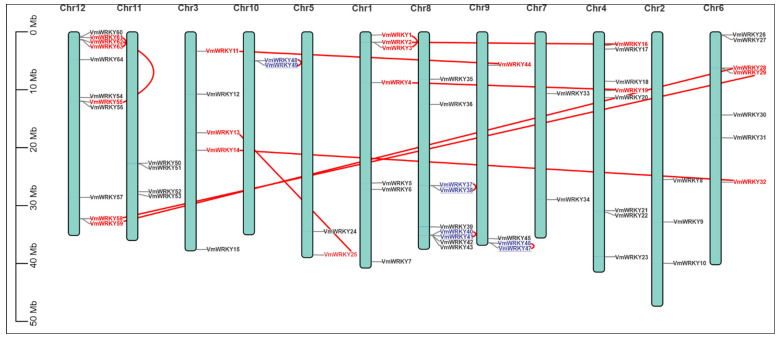
Chromosomal map and coordinates of *VmWRKY* gene duplication events: The identity of each linkage group is indicated in the central part of each bar. The putative segmental duplicated genes are connected by red color lines, and the duplicated gene pair in tandem is represented by the blue color on the chromosome.

**Figure 3 plants-12-03176-f003:**
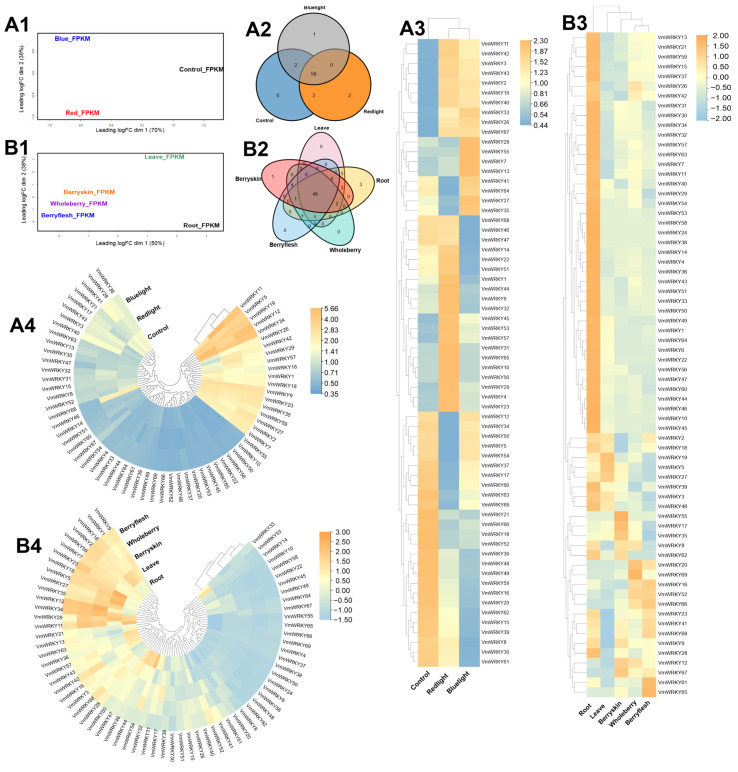
(**A**) Gene expression of bilberry transcriptomic light samples (under control, red light, and blue light conditions, measured in FPKM): (1) multi-dimensional scaling (MDS) plot of the light samples; (2) *VmWRKY* gene expression common to each sample shown in a Venn diagram; (3) differential expression of *VmWRKY* genes in each sample (values presented in log2, ranging from 0.40 to 2.30); (4) highly expressed *VmWRKY* genes in all samples (values presented in log2, ranging from 0.35 to 5.66). (**B**) Gene expression of bilberry transcriptomic organ samples (in leaves, roots, whole berries, berry flesh, and berry skin, measured in FPKM): (1) multi-dimensional scaling (MDS) plot of the organ samples (roots, leaves, berry skin, whole berry, and berry flesh); (2) *VmWRKY* gene expression common to each sample shown in a Venn diagram; (3) differential expression of *VmWRKY* genes in each sample (values presented in log2, ranging from −2.00 to 2.00); (4) highly expressed *VmWRKY* genes in all samples (values presented in log2, ranging from −1.50 to 3.00).

**Table 1 plants-12-03176-t001:** Information on *VmWRKY* genes in the bilberry genome (*Vaccinium myrtillus* L.).

Gene Name	Gene Identifier	Group	Chr.	Start	End	Strand	CDS	(aa)	MW	pI	I. Index	GRAVY
Vmy01g15360	*VmWRKY1*	I	Chr1	562,585	565,972	+	2028	675	73,184.1101	5.8614	57.63496	−0.78104
Vmy01g15494	*VmWRKY2*	I	Chr1	1,805,033	1,810,979	−	1704	567	61,989.0022	6.7926	52.19947	−0.75467
Vmy01g15496	*VmWRKY3*	I	Chr1	1,815,400	1,821,385	−	1356	451	48,425.053	4.7083	62.64989	−0.84169
Vmy01g16077	*VmWRKY4*	IIb	Chr1	8,788,109	8,792,213	+	1617	538	58,725.4997	6.5958	45.7948	−0.70985
Vmy01g17102	*VmWRKY5*	I	Chr1	26,099,661	26,103,183	+	1425	474	51,364.8663	8.9407	45.66751	−0.80696
Vmy01g17169	*VmWRKY6*	Iic	Chr1	27,189,670	27,191,923	−	876	291	32,429.5196	5.0955	58.33574	−0.73814
Vmy01g18042	*VmWRKY7*	Iic	Chr1	39,683,976	39,685,598	+	1017	338	37,888.7482	5.9652	61.04822	−0.88817
Vmy02g31145	*VmWRKY8*	Iic	Chr2	25,519,147	25,521,976	+	876	291	32,444.1819	6.4528	70.26117	−1.01993
Vmy02g31720	*VmWRKY9*	I	Chr2	32,825,778	32,832,842	+	1611	536	58,052.007	6.6116	58.72351	−0.81287
Vmy02g32314	*VmWRKY10*	Iib	Chr2	39,960,782	39,963,444	+	1578	525	57,472.7052	5.2647	48.67124	−0.84019
Vmy03g6652	*VmWRKY11*	Iia	Chr3	3,347,599	3,349,848	+	1008	335	37,183.4869	7.5707	54.8606	−0.66806
Vmy03g7279	*VmWRKY12*	Iic	Chr3	10,749,229	10,756,210	−	807	268	29,564.1446	5.4622	62.2306	−0.84179
Vmy03g7725	*VmWRKY13*	Iic	Chr3	17,418,681	17,421,414	−	537	178	20,532.6635	9.6275	33.60225	−1.04831
Vmy03g7865	*VmWRKY14*	Iie	Chr3	20,460,927	20,465,147	−	867	287	32,027.1993	5.1769	64.9216	−0.81986
Vmy03g9094	*VmWRKY15*	Iic	Chr3	37,596,334	37,598,020	+	906	301	33,881.2384	5.6614	65.33953	−0.84153
Vmy04g26811	*VmWRKY16*	I	Chr4	2,341,763	2,347,090	−	1782	593	64,536.4045	6.4074	51.08398	−0.73693
Vmy04g26879	*VmWRKY17*	Iib	Chr4	2,979,626	2,984,279	+	1467	488	53,622.5498	9.0454	59.18258	−0.72992
Vmy04g27403	*VmWRKY18*	I	Chr4	8,533,619	8,536,143	−	1437	478	52,408.7018	6.7441	55.2864	−1.03536
Vmy04g27528	*VmWRKY19*	Iib	Chr4	10,145,599	10,147,984	+	1575	524	57,082.0727	6.3267	45.18626	−0.71164
Vmy04g27619	*VmWRKY20*	Iie	Chr4	11,366,863	11,367,486	−	624	207	23,294.8101	6.5994	60.09614	−0.81063
Vmy04g28682	*VmWRKY21*	Iic	Chr4	30,876,942	30,884,972	−	843	280	31,697.0249	6.3051	53.21964	−0.76571
Vmy04g28697	*VmWRKY22*	Iic	Chr4	31,197,907	31,202,208	+	564	187	21,115.9108	5.7896	58.50588	−0.97701
Vmy04g29204	*VmWRKY23*	I	Chr4	38,844,924	38,855,142	+	384	513	56,570.0032	5.5428	53.90975	−0.86667
Vmy05g13713	*VmWRKY24*	Iib	Chr5	34,491,956	34,498,576	−	1734	577	63,130.9269	6.6639	50.274	−0.78943
Vmy05g14091	*VmWRKY25*	Iic	Chr5	38,520,966	38,524,762	−	396	131	15,231.1171	9.7211	37.99313	−1.08397
Vmy06g33434	*VmWRKY26*	Iic	Chr6	535,034	539,542	−	798	265	29,556.1168	9.1886	62.50906	−0.74113
Vmy06g33447	*VmWRKY27*	Iie	Chr6	641,618	643,258	−	978	325	37,054.0126	4.6838	76.58988	−1.212
Vmy06g34052	*VmWRKY28*	Iid	Chr6	6,251,528	6262,173	−	1512	503	57,035.1503	9.8303	46.33002	−0.57475
Vmy06g34072	*VmWRKY29*	I	Chr6	6,434,377	6,437,276	+	1716	571	62,463.3081	7.7103	65.48984	−0.9387
Vmy06g34731	*VmWRKY30*	Iie	Chr6	14,371,009	14,374,552	−	1452	483	52,681.3465	5.6884	53.77598	−0.87433
Vmy06g35042	*VmWRKY31*	Iib	Chr6	18,292,323	18,295,336	+	1761	586	63,759.1384	6.3526	49.9041	−0.69369
Vmy06g35510	*VmWRKY32*	Iie	Chr6	25,953,649	25,954,811	+	816	271	30,225.875	5.2132	58.61476	−0.86236
Vmy07g24811	*VmWRKY33*	Iie	Chr7	10,670,375	10,672,822	−	1017	338	37,204.4243	5.2683	72.40296	−0.86982
Vmy07g25848	*VmWRKY34*	Iid	Chr7	28,979,073	28980568	+	1038	345	37,872.4647	9.7013	44.33942	−0.63072
Vmy08g19321	*VmWRKY35*	I	Chr8	8,189,304	8,193,703	−	1566	521	57,275.1267	7.6594	53.98618	−0.78061
Vmy08g19634	*VmWRKY36*	Iid	Chr8	12,539,067	12,540,851	+	975	324	35,083.2628	9.5829	49.67099	−0.5071
Vmy08g20363	*VmWRKY37*	Iib	Chr8	26,539,533	26,543,458	−	1257	418	46,326.055	6.7756	39.74285	−0.76555
Vmy08g20364	*VmWRKY38*	Iib	Chr8	26,567,087	26,572,765	−	1314	437	48,350.2147	8.0178	44.91215	−0.78558
Vmy08g20862	*VmWRKY39*	III	Chr8	33,668,487	33,685,457	+	2013	670	75,108.3868	6.8035	52.55225	−0.57209
Vmy08g20951	*VmWRKY40*	III	Chr8	35,084,050	35,087,270	+	726	241	27,218.4897	9.0240	45.36892	−0.64398
Vmy08g20953	*VmWRKY41*	III	Chr8	35,124,873	35,128,367	+	726	241	27,276.5258	8.8811	45.29091	−0.66598
Vmy08g20957	*VmWRKY42*	III	Chr8	35,152,034	35,152,550	−	336	111	13,001.3469	6.8845	33.74234	−0.95405
Vmy08g20958	*VmWRKY43*	III	Chr8	35,155,907	35,158,286	−	351	116	13,315.6492	6.3954	48.45517	−0.75517
Vmy09g21848	*VmWRKY44*	Iia	Chr9	5,768,052	5,770,593	+	987	328	36,404.0856	7.6022	48.30549	−0.76738
Vmy09g23773	*VmWRKY45*	I	Chr9	35,699,795	35,704,437	−	1659	552	61,427.4531	7.1727	59.53261	−0.97409
Vmy09g23833	*VmWRKY46*	III	Chr9	36,446,627	36,448,661	−	1008	335	37,488.9147	5.4796	57.01851	−0.78657
Vmy09g23834	*VmWRKY47*	III	Chr9	36,456,426	36,458,322	−	981	326	36,383.8599	5.9442	56.7635	−0.7227
Vmy10g9485	*VmWRKY48*	Iia	Chr10	4,985,336	4,988,127	−	870	289	32,320.8629	6.3253	39.20208	−0.76332
Vmy10g9486	*VmWRKY49*	Iia	Chr10	5,039,428	5,040,970	+	540	179	19,928.3004	8.6561	41.61508	−0.54078
Vmy11g4858	*VmWRKY50*	III	Chr11	22,694,878	22,709,418	+	1887	628	68,221.588	5.7820	60.61369	−0.63392
Vmy11g4864	*VmWRKY51*	III	Chr11	22,834,006	22,838,063	−	951	316	35,353.069	6.0694	57.9981	−0.70475
Vmy11g5266	*VmWRKY52*	Iie	Chr11	27,599,930	27,601,824	+	1050	349	37,869.319	5.2802	60.43413	−0.75387
Vmy11g5322	*VmWRKY53*	III	Chr11	28,097,858	28,099,773	−	1152	383	42,063.9735	5.9608	54.54885	−0.72298
Vmy12g1272	*VmWRKY54*	III	Chr12	11,342,519	11,344,263	+	1071	356	40,375.9677	5.5414	46.38343	−0.88736
Vmy12g1322	*VmWRKY55*	Iic	Chr12	11,966,502	11,968,411	+	498	165	18,670.9435	9.8000	32.01515	−0.85818
Vmy12g1324	*VmWRKY56*	Iie	Chr12	11,987,949	11,989,528	−	957	318	34,707.0416	5.4257	54.09343	−0.53899
Vmy12g2389	*VmWRKY57*	I	Chr12	28,589,923	28,593,312	−	1644	547	60,328.8928	8.4261	56.26289	−0.9947
Vmy12g2722	*VmWRKY58*	I	Chr12	32,281,670	32,282,541	+	693	230	25,178.2666	6.4438	66.10478	−0.94739
Vmy12g2723	*VmWRKY59*	Iid	Chr12	32,286,557	32,288,334	+	1023	340	38,242.1155	9.7851	58.70412	−0.78235
Vmy12g371	*VmWRKY60*	III	Chr12	776,559	778,122	−	963	320	36,043.8189	4.8940	38.90625	−0.60844
Vmy12g391	*VmWRKY61*	Iie	Chr12	961,923	971,203	−	2223	740	82,707.5643	4.9592	53.45962	−0.44878
Vmy12g427	*VmWRKY62*	Iic	Chr12	1,332,136	1,333,788	−	597	198	22,659.3469	9.1573	49.19899	−0.82424
Vmy12g435	*VmWRKY63*	Iie	Chr12	1,390,928	1,392,563	−	969	322	35,739.1232	5.4169	61.10155	−0.85621
Vmy12g744	*VmWRKY64*	Iic	Chr12	4,792,418	4,798,547	−	708	235	26,134.3628	7.1288	50.72255	−0.97064
VmyS4054g6302	*VmWRKY65*	Iib	4054	11,361	14,074	+	1164	387	41,562.6356	8.9565	47.0261	−0.54419
VmyS5938g36257	*VmWRKY66*	I	5938	264,671	269,317	+	1128	375	41,777.2506	6.9077	44.0112	−0.72293
VmyS6208g149	*VmWRKY67*	Iic	6208	30,683	48,873	−	1812	603	66,813.4835	9.3106	43.03997	−0.62919
VmyS7930g26553	*VmWRKY68*	IV	7930	47,310	57,551	+	1482	493	56,297.4289	9.0159	54.64016	−0.52312
VmyS8810g36160	*VmWRKY69*	IV	8810	23,416	30,743	−	1170	389	43,454.744	5.6507	54.54267	−0.70103

aa = Amino acid; pI = isoelectric point; MW = molecular weight; GRAVY = large average hydrophobicity; I. index = instability index.

**Table 2 plants-12-03176-t002:** Ks, Ka, and Ka/Ks calculation and divergence time of duplicated bilberry *VmWRKY* gene pairs.

Duplicate Gene Pair	Ka	Ks	Ka_Ks	Duplicated Type	Time (Mya) *
*VmWRKY37*/*VmWRKY38*	NaN	NaN	NaN	Tandem	−
*VmWRKY40/VmWRKY41*	0.0018229	0.0116	0.1571441	Tandem	0.138938872
*VmWRKY46/VmWRKY47*	NaN	NaN	NaN	Tandem	−
*VmWRKY48/VmWRKY49*	0.2153629	0.4347175	0.4954089	Tandem	16.41485794
*VmWRKY4/VmWRKY19*	0.1863388	1.2026591	0.154939	Segmental	14.20264977
*VmWRKY65/VmWRKY24*	0.0117773	0.1173245	0.1003819	Segmental	0.897657317
*VmWRKY11/VmWRKY44*	0.1737659	0.7309832	0.2377154	Segmental	13.24435556
*VmWRKY14/VmWRKY32*	3.34641 **	4.13312 **	0.80965 **	Segmental	3.5265 **
*VmWRKY61/VmWRKY63*	1.0942903	1.7098092	0.6400073	Segmental	83.40627188
*VmWRKY59/VmWRKY28*	NaN	NaN	NaN	Segmental	−
*VmWRKY67/VmWRKY22*	0.9085855	NaN	NaN	Segmental	69.25193994
*VmWRKY13/VmWRKY25*	NaN	NaN	NaN	Segmental	−
*VmWRKY62/VmWRKY55*	NaN	NaN	NaN	Segmental	−
*VmWRKY2/VmWRKY16*	NaN	NaN	NaN	Segmental	−
*VmWRKY1/VmWRKY3*	0.006383	0.0427162	0.1494286	Segmental	0.486510442
*VmWRKY58/VmWRKY29*	0	NaN	NaN	Segmental	0

* Mya, Million years ago. ** The values were computed using the KaKs_Calculator2.0 program, applying the Nei and Gojobori (NG) method. NaN = Not a number (refers to an undefined value or a result that cannot be calculated).

## Data Availability

Not applicable.

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
