# Peer review of "Genome-Wide Identification of Bilberry WRKY Transcription Factors: Go Wild and Duplicate"

_plants, 2023, doi:10.3390/plants12183176_

Round 1

Reviewer 1 Report

In this manuscript, Felipez and collaborators perform a genome-wide identification and characterization of WRKY transcription factors in bilberry using several bioinformatic approaches. The approach used is similar to other publications dealing with in silico identification of several protein groups in which they add RNAseq support using publicly available data. The results can be of interest for researchers of the field, therefore the manuscript would be suitable for publication.

However there are some issues that need to be solved and general improvements that could be added before acceptance. The material and method section need some improvement is different sections, and similarly the discussion could be a bit enhanced.

First of all, I ask the authors to thoroughly revise the citations, as there are a number of inconsistencies (for instance, the bilberry reference genome is Wu et al, 2021, not Samkumar et al, 2021. The role of WRKY TFs in anthocyanin regulation could be a bit emphasised in the introduction, since it is a focus of the paper (see for instance Peng et al, 2020 and others).

Another issue in the introduction is the apparent incorrect citation of "blueberries"-related papers. For instance, in line 44 they write about V. corymbosum, but references describe V. myrtillus (2) and V. macrocarpon (9), and similar inexact citations follow in lines 47-48. I would rephrase also the sentence in line 45, since raw data are actually available, assembled transcripts, viceversa, are not. On the other side, similar data have been made available by Thole et al (2019) and Rowland et al, 2012, etc. Moreover, since genomic (and some transcriptomic) data are indeed available for download (https://www.vaccinium.org/data_download), it is not clear to me what the authors intend to say.

Line 109 and following: the different transcripts could actually represent different isoforms due to alternative splicing events?

Line 122 and following: the identification of sequences recalling cis-regulatory elements does not necessarily imply they are de facto (this would need to be proven), so I would be cautious and maybe insert a “putative” in the phrase.

Section 2.5 (line 165): I ask the authors, in the related section of Materials and methids, to explain what they mean (ie, what is the threshold, if any, they applied) to distinguish genes between expressed and not expressed.

It is not clear to me what the authors mean in “The differential expression of VmWRKY genes in samples of anthocyanin biosynthesis regulation by abscisic acid” (line 175): do they identify genes regulated by ABA or they used a library where the samples were treated with the phytohormone? In the latter case, could they find a correlation in samples with a related cis-element (ABA, light-related)? This would enforce section 2.2.

Section 3.5: the value 24,491 reported in line 357 (similar cases downstream) are referred to what? Total expression of VmWRKY genes or the number of expressed genes? In general this section should be carefully checked (ex.: line 366: you cannot compare different experiments in different species using different protocols). Light induce anthocyanins expression, but different wavelengths behave differently: check Tao et al, 2018, Planta for instance; can you see differences in bilberry? There is space for improving the section.

Section 4.5: please define better how you identified segmental duplications as well as tandem duplications

Line 440: EdgeR (correct the typo) and DESeq2 are not used for quantification but for differential expression, and they need counts (such as those obtained by FeatureCounts), not FPKM-normalized values: those may be used to draw heatmaps.

In the conclusions maybe some lines about the importance of what was charaterized in the paper would be useful to stress the importance of the paper. Here and in the introduction you may benefit of what Edger and coauthors write in 2022 in Horticulture Research.

Figure 2: blue colour is not really identifiable; please consider a change (underlining maybe?).

Figure 3: please explain what the error bars indicate. How is it possible that all samples show the same pattern in panel D?

Table 2: authors should explain what NaN mean in this context and, in materials and methods, how was this calculation performed.

please revise the manuscript for better clarity in some parts

Author Response

 Dear Reviewer #1,

Thank you for considering our manuscript for publication.

We are sure that this study will improve

1) First of all, I ask the authors to thoroughly revise the citations, as there are a number of inconsistencies (for instance, the bilberry reference genome is Wu et al, 2021, not Samkumar et al, 2021.

Answer: The authors reviewed and agreed with the comments and observations of the reviewer. However, the methodology focuses on referencing Wu et al., 2021 [4] as the genome used in the in-silico research.

2) The role of WRKY TFs in anthocyanin regulation could be a bit emphasized in the introduction, since it is a focus of the paper (see for instance Peng et al, 2020 and others).

Answer: The authors agreed to focus on the role of FT WRKYs in anthocyanin biosynthesis in the introduction. Therefore, the role of WRKYs in biotic and abiotic stress was removed, and reports on FT WRKYs in anthocyanin biosynthesis were included as referenced in the review by Felipez et al., 2022.

3) Another issue in the introduction is the apparent incorrect citation of "blueberries"-related papers. For instance, in line 44 they write about V. corymbosum, but references describe V. myrtillus (2) and V. macrocarpon (9), and similar inexact citations follow in lines 47-48.

Answer: The authors agree with the reviewer's suggestions, which included reviewing the references and removing inconsistent citations. Furthermore, the text was rewritten to better understand the importance of studying the bilberry (V. myrtillus).

4) Line 109 and following: the different transcripts could actually represent different isoforms due to alternative splicing events?

Answer: We agree with the question. In fact, alternative splicing is an important mechanism to increase the diversity of proteins in the organism and plays a crucial role in the regulation of gene expression and in the response to different physiological and pathological conditions. However, we performed a detailed analysis using Smith-Waterman local alignment, where the percentage of amino acid sequence similarity was greater than 98% (Supplement 1). Additionally, we also examined the structure of the motives and domains presented in Figure 1 and no whole domain is missing. Therefore, considering these aspects, the analyzes were clarified in the manuscript.

5) Line 122 and following: the identification of sequences recalling cis-regulatory elements does not necessarily imply they are de facto (this would need to be proven), so I would be cautious and maybe insert a “putative” in the phrase.

Answer: The term "putative" was added as a complement.

6) Section 2.5 (line 165): I ask the authors, in the related section of Materials and methods, to explain what they mean (ie, what is the threshold, if any, they applied) to distinguish genes between expressed and not expressed

Answer: It was supplemented and clarified in the methodology. However, in the analysis of gene expression in each transcriptomic sample, the proportion was calculated based on the total annotated genes in the bilberry reference genome (36,405 genes = 100%).

7) It is not clear to me what the authors mean in “The differential expression of VmWRKY genes in samples of anthocyanin biosynthesis regulation by abscisic acid” (line 175): do they identify genes regulated by ABA or they used a library where the samples were treated with the phytohormone? In the latter case, could they find a correlation in samples with a related cis-element (ABA, light-related)? This would enforce section 2.2.

Answer: The authors agree with the observation made in the writing of this section regarding the regulation of anthocyanin by phytohormones (ABA). This information has been removed and clarified, emphasizing the expression of WRKY genes in light-induced transcriptomic samples (PRJNA739815) and their involvement in anthocyanin biosynthesis. However, it is important to note that phytohormones (ABA) are also involved in the induction of anthocyanin biosynthesis, which has been complemented in the discussion section.

8) Section 3.5: the value 24,491 reported in line 357 (similar asesdownstream) are referred to what? Total expression of VmWRKY genes or the number of expressed genes? In general, this section should be carefully checked (ex.: line 366: you cannot compare different experiments in different species using different protocols).

Answer: We agree that it is not clear. However, more details have been added to the methodology of transcriptomic analysis, specifically regarding the processes of the following steps. Reference was made to the proportion of expression in the transcriptomic samples relative to the total annotated genes in the blueberry reference genome (36,405 genes = 100%).

9) Light induces anthocyanins expression, but different wavelengths behave differently: check Tao et al, 2018, Planta for instance; can you see differences in bilberry? There is space for improving the section.

Answer: The authors agree with the suggestions. However, it is important to clarify that the article "The blue light signal transduction pathway is involved in anthocyanin accumulation in 'Red Zaosu' pear" describes the induction of anthocyanin biosynthesis by blue light compared to red light in pear fruit. Although this article is relevant, it is important to note that there are other genes involved in the induction of anthocyanin biosynthesis, which could lead to confusion in the manuscript. This analysis in underway for another manuscript. To avoid misunderstandings, these genes have been duly addressed and discussed in the introduction and discussion sections.

10) Section 4.5: please define better how you identified segmental duplications as well as tandem duplications Line 440: EdgeR (correct the typo) and DESeq2 are not used for quantification but for differential expression, and they need counts (such as those obtained by FeatureCounts), not FPKMnormalized values: those may be used to draw heatmaps.

Answer1: Additional details were added to the methodology regarding the criteria used for identifying tandem and segmental duplications in section 4.5, as well as the genetic divergence of the duplicated genes.

Answer 2: We agree with the reviewer regarding the packages used in R Studio, EdgeR, and DESeq2. They were used for the analysis of gene expression quantification, not for the actual quantification of the total count of aligned reads (using the FeatureCounts library). This clarification was provided in section 2.6 of the manuscript.

11) In the conclusions maybe some lines about the importance of what was characterized in the paper would be useful to stress the importance of the paper.

Answer: In section 5, additional details have been added regarding the importance of characterizing VmWRKY genes, phylogenetic relationships, and differential expressions, as well as their involvement in anthocyanin biosynthesis induction.

12) Here and in the introduction, you may benefit of what Edger and coauthors write in 2022 in Horticulture Research.

Answer: A paragraph with reference to Edger 2022 has been added to the introduction.

13) Figure 2: blue colour is not really identifiable; please consider a change (underlining maybe?).

      Answer: It has been improved and corrected.

14) Figure 3: please explain what the error bars indicate. How is it possible that all samples show the same pattern in panel D?.

Answer:  We agree with the observation; however, the following description could provide clarity:

  1. a) In graphs 3B1-3B5 and 3D1-3D5, the genes highly expressed in all samples were shown, the differential expression in each sample was not shown. For that matter, the error bars capture cross-replication variability and measurement uncertainty as estimated by the Cuffdiff RNA-seq statistical model.

  1. b) To provide clarity in the analysis, we have graphed and analyzed this section, focusing on the distribution of expression similarity or dissimilarity samples, common expressions of VmWRKY genes in the samples, differential expression and highly expressed genes in all samples. in addition to providing complementary analysis 5.

15) Table 2: authors should explain what NaN mean in this context and, in materials and methods, how was this calculation performed.

Answer: What is NaN was added in the same table 2, however, they were analyzed again in different formats and programs, using genomic sequences of DNA, CDS, amino acids, in addition to the genomic annotation of the gff file, as required for you calculates, all forced all forced that not being able to do the calculation, because the sequences are implements. This calculation error is probably due to the information available in the blueberry genome assembly and annotation. By trying other forms of calculation, only one additional pair of duplicated genes could have their divergence dated. This was added to the results.

Reviewer 2 Report

In the manuscript entitled “Genome-wide identification of Bilberry WRKY transcription factors: gene pairs and stress responses” the authors identify WRKY transcription factors of bilberry obtained from a published genome in the Genome Database of Vaccinium and provide a functional characterization of them based on gene structure and domain conservation. They also study the duplication events of VmWRKY genes found along the chromosomes and show gene expression data obtained from two different RNA-seq databases deposited at NCBI.

Due to the benefits of anthocyanins in human health, there is an increasing interest on anthocyanin-rich fruits. For this reason, great efforts have been done in the last years to decipher the genetic/molecular/metabolic mechanisms that undergo anthocyanin biosynthesis in plants of the vaccinium genera (blueberries, cranberries, bilberries, etc.). Indeed, a genome database is available for public research (GDV).

As WRKY transcription factors are known to regulate plant responses to biotic and abiotic stresses and they also mediate in the regulation of anthocyanin biosynthesis this paper may contribute to increase knowledge in this plant species.

In general terms the analyzes provided in the manuscript have been appropriately described, although several aspects could be improved. On the other hand, the results presented in this manuscript rely entirely on in silico data obtained from public datasets, but some biological assays that complement/validate the in silico results (common in related papers) are lacking. For this reason, I consider that data could be further exploited in order to increase suitable information and interest.

Furthermore, the gene expression analyzes described in the manuscript include samples of bilberry plants grown under different sources of light and also samples obtained from different bilberry plant tissues; although there are common plant responses integrating both light perception and stress responses, the manuscript do not include any specific experiment related to stress (biotic/abiotic stress conditions or application of stress-related hormones), for this reason the term “stress responses” should be removed from the title, as it is confusing. If the authors are interested in including stress data there is available a public dataset (PRJNA481170) containing the transcriptional profiling of bilberry plants treated with MeJA.

Abstract:

Check the wording of the text in paragraph (lines 23-24) as it is confusing.

1.     Introduction:

                  To date there is an extensive bibliography regarding the influence of different light intensities in gene expression, growth rate and anthocyanin accumulation in several species of vaccinium (e.g. Zhang et al., 2022 (DOI 10.3389/fpls.2022.1073332); Guo et al., 2022 (https://doi.org/10.1186/s12870-022-03585-x); Wu et al., 2021 (DOI: 10.1111/1755-0998.13467)). Some of them have been mentioned in the manuscript in other paragraphs (Samkumar et al., 2021 (DOI: 10.1111/pce.14158)), but as the expression profiles of VmWRKY genes under different light conditions are being displayed in the manuscript, a deeper explanation of the importance of light should be included in the introduction.

                  Re-write the paragraph contained in lines 84-86 as it is confusing.

2.     Results:

2.1   Identification of the WRKY protein family

Line 112. I cannot find Supplementary S1 in the provided supplementary file.

2.2 Analysis of cis elements in WRKY gene promoters

A table (or graph) should be included providing relevant information such as frequency of cis-elements and a brief description of the biological pathways they are involved in (hormone responses, stress responses, metabolite biosynthesis, etc.)

Line 130. I cannot find Supplementary S2 in the provided supplementary file.

2.3 Phylogeny, gene structure and motif analysis of WRKY proteins

Figure 1.

-        A brief description of the motifs could be included:

§  Identify conserved regions (WRKY elements, Leucine-zipper, zinc-finger, etc.)

§  Show an average sequence (or a frequency diagram) of other conserved regions that cannot be identified.

-         Arabidopsis thaliana orthologs could also be included in the phylogenetic tree. It would help identify the closest orthologs of each gene and allow to go deeper in knowledge through bibliography if desired. You may see Baillo et al., 2020 (https://doi.org/10.1371/ journal.pone.0236651) as an example.

-        Line 210: UTRs are indicated by blue boxes (instead of green).

Table 2: Additional information such as the conserved heptapeptide class and the zinc-finger type of the protein could be included.

2.5 Number of transcripts and expression patterns of VmWRKY genes

- Line 166, 167, 176, 228…: “the transcriptional expression of genes for each sample of abscisic acid-regulated anthocyanin biosynthesis…” is not an appropriate description of the samples in the study. The samples come from the experiments described in Samkumar et al., 2021 (DOI: 10.1111/pce.14158) where light quality produced differences in the expression of some ABA and anthocyanin biosynthetic genes. This description is confusing and it should be modified.

- A table identifying those genes that are common to every condition or unique should be added.

- Line 174. “Supplementary S4”?? or Supplementary S5. Figure 2??

- Paragraph 175-183: It is confusing and should be re-written.

- In my opinion the expression plots depicted in Figure 3 B and D do not offer extra information from what is represented in the heatmaps (Figure A and B). Sample variability could be also visualized including individual samples, instead of the mean of expression between samples, in the heatmap.

-Genes represented in Figure 3B do not seem to show “CLEAR differences in expression in conditions control, red and blue light” (line 229), and they are not relevant as light-sensitive genes at all.

In any case, what seems more important to me is the relative accumulation of each sequence compared to the control (differential expression) and not only if it is present or not in a sample. The ratio of expression should also be considered: genes with a significant differential expression between control and red/blue light conditions should be highlighted and identified in a separate table (e.g., VmWRKY42, VmWRKY58, VmWRKY43)

-On the other hand, there is no correlation between the FPKM values in the bar diagrams in Figure 3D and the expression levels represented in the heatmap (Figure 3C). Probably it is because the sample order (left to right) is also different. Furthermore, the error bars in the plots seem to be wrong.

In my opinion the bar plots in Figure 3 could be removed. However, it could be interesting to show gene correlation between heatmaps instead, as one would expect that light-sensitive genes should show differential expression in different plant organs. For example, a co-expression network correlating light-sensitive genes with genes differentially expressed in plant organs; or a new clustering tree could be displayed integrating all 8 samples for light-sensitive WRKY genes.

3.     Discussion:

3.2 Analysis of cis elements in VmWRKY gene promoters.

- Is there a correlation between WRKY groups and the nature/frequency of different cis-regulatory elements? It is particularly relevant for light-responsive elements (see paragraph 270-273) and it could be informative as it could allow establish a relationship between light-mediated transcriptional regulation of WRKY genes and protein functional domains. It could be an interesting result to show.

3.3 Phylogeny, gene structure and motif analysis of WRKY proteins.

- Paragraph 294-299: The numbers assigned to each group do no coincide with those showed in Figure 1. Furthermore, it seems to be redundant, as it has been previously explained in results.

- Line 301. Please, use “abundant” instead of “expressive”.

3.5 Phylogeny, gene structure and motif analysis of WRKY proteins.

- Paragraph 355-357. It has no sense to me. Furthermore, the authors keep on using the wrong expression “abscisic acid-regulated anthocyanin biosynthesis”.

4.     Materials and Methods:

4.6 Transcriptomic analysis of VmWRKY genes

- Line 425. Wrong reference: As far as I know, experiment PRJNA739815 is described in Wu et al., 2021(DOI: 10.1111/1755-0998.13467)

- Line 430: Wrong reference: Experiment PRJNA747684 is described in Samkumar et al., 2021 (DOI: 10.1111/pce.14158)

- Lines 432,433: Include the correct sample names for red and blue light experiments

Supplementary files:

                  Most of the supplementary figures provided with the manuscript (Supplementary 4. Figures 1, 2 and 3; Supplementary 5. Figures 2 and “also 2”) are not referenced. The only one that seems to be included in the manuscript is “Supplementary 5. Figure 2” (one of them), appearing in line 174 as “Supplementary S4”.

                  Line 112. “Supplementary S1” is lacking in supplementary files.

                  Line 130. “Supplementary S2” is lacking in supplementary files.

                  A complete and revised file of supplementary figures should be added before publication.

Author Response

Reviewer #2:

In general terms the analyzes provided in the manuscript have been appropriately described, although several aspects could be improved. On the other hand, the results presented in this manuscript rely entirely on in silico data obtained from public datasets, but some biological assays that complement/validate the in silico results (common in related papers) are lacking. For this reason, I consider that data could be further exploited in order to increase suitable information and interest.

Comments:                                      

  • To date there is an extensive bibliography regarding the influence of different light intensities in gene expression, growth rate and anthocyanin accumulation in several species of vaccinium (e.g. Zhang et al., 2022 (DOI 10.3389/fpls.2022.1073332); Guo et al., 2022 (https://doi.org/10.1186/s12870-022-03585-x); Wu et al., 2021(DOI: 10.1111/1755-0998.13467).
  • Some of them have been mentioned in the manuscript in other paragraphs (Samkumar et al., 2021 (DOI: 10.1111/pce.14158)), but as the expression profiles of VmWRKY genes under different light conditions are being displayed in the manuscript, a deeper explanation of the importance of light should be included in the introduction.
  • Re-write the paragraph contained in lines 84-86 as it is confusing.
  • 1 Identification of the WRKY protein family Line 112. I cannot find Supplementary S1 in the provided supplementary file.
  • 2 Analysis of cis elements in WRKY gene promoters A table (or graph) should be included providing relevant information such as frequency of ciselements and a brief description of the biological pathways they are involved in (hormone responses, stress responses, metabolite biosynthesis, etc.) Line 130. I cannot find Supplementary S2 in the provided supplementary file.
  • 3 Phylogeny, gene structure and motif analysis of WRKY proteins Figure 1. - A brief description of the motifs could be included: § Identify conserved regions (WRKY elements, Leucine-zipper, zinc-finger, etc.) § Show an average sequence (or a frequency diagram) of other conserved regions that cannot be identified.
  • Arabidopsis thaliana orthologs could also be included in the phylogenetic tree. It would help identify the closest orthologs of each gene and allow to go deeper in knowledge through bibliography if desired. You may see Baillo et al., 2020 (https://doi.org/10.1371/ journal.pone.0236651) as an example. - Line 210: UTRs are indicated by blue boxes (instead of green).
  • Table 2: Additional information such as the conserved heptapeptide class and the zinc-finger type of the protein could be included.
  • 5 Number of transcripts and expression patterns of VmWRKY genes - Line 166, 167, 176, 228…: “the transcriptional expression of genes for each sample of abscisic acid-regulated anthocyanin biosynthesis…” is not an appropriate description of the samples in the study.
  • The samples come from the experiments described   in Samkumar et al., 2021 (DOI: 10.1111/pce.14158) where light quality produced differences in the expression of some ABA and anthocyanin biosynthetic genes. This description is confusing and it should be modified.
  • This description is confusing and it should be modified. - A table identifying those genes that are common to every condition or unique should be added. - Line 174. “Supplementary S4”?? or Supplementary S5. Figure 2?? - Paragraph 175-183: It is confusing and should be re-written. - In my opinion the expression plots depicted in Figure 3 B and D do not offer extra information from what is represented in the heatmaps (Figure A and B).
  • Sample variability could be also visualized including individual samples, instead of the mean of expression between samples, in the heatmap. -Genes represented in Figure 3B do not seem to show “CLEAR differences in expression in conditions control, red and blue light” (line 229), and they are not relevant as light-sensitive genes at all.
  • In any case, what seems more important to me is the relative accumulation of each sequence compared to the control (differential expression) and not only if it is present or not in a sample. The ratio of expression should also be considered: genes with a significant differential expression between control and red/blue light conditions should be highlighted and identified in a separate table (e.g., VmWRKY42, VmWRKY58, VmWRKY43).
  • On the other hand, there is no correlation between the FPKM values in the bar diagrams in Figure 3D and the expression levels represented in the heatmap (Figure 3C). Probably it is because the sample order (left to right) is also different. Furthermore, the error bars in the plots seem to be wrong.
  • In my opinion the bar plots in Figure 3 could be removed. However, it could be interesting to show gene correlation between heatmaps instead, as one would expect that light-sensitive genes should show differential expression in different plant organs. For example, a co-expression network correlating lightsensitive genes with genes differentially expressed in plant organs; or a new clustering tree could be displayed integrating all 8 samples for light-sensitive WRKY genes.
  • 2 Analysis of cis elements in VmWRKY gene promoters. - Is there a correlation between WRKY groups and the nature/frequency of different cis-regulatory elements? It is particularly relevant for lightresponsive elements (see paragraph 270-273) and it could be informative as it could allow establish a relationship between light-mediated transcriptional regulation of WRKY genes and protein functional domains. It could be an interesting result to show.
  • 3 Phylogeny, gene structure and motif analysis of WRKY proteins. - Paragraph 294-299: The numbers assigned to each group do no coincide with those showed in Figure 1. Furthermore, it seems to be redundant, as it has been previously explained in results. - Line 301. Please, use “abundant” instead of “expressive”.
  • 5 Phylogeny, gene structure and motif analysis of WRKY proteins. - Paragraph 355-357. It has no sense to me. Furthermore, the authors keep on using the wrong expression “abscisic acid-regulated anthocyanin biosynthesis”.
  • Materials and Methods: 4.6 Transcriptomic analysis of VmWRKY genes - Line 425. Wrong reference: As far as I know, experiment PRJNA739815 is described in Wu et al., 2021(DOI: 10.1111/1755-0998.13467) - Line 430: Wrong reference: Experiment PRJNA747684 is described in Samkumar et al., 2021 (DOI: 10.1111/pce.14158) - Lines 432,433: Include the correct sample names for red and blue light experiments.
  • Supplementary files: Most of the supplementary figures provided with the manuscript (Supplementary 4. Figures 1, 2 and 3; Supplementary 5. Figures 2 and “also 2”) are not referenced. The only one that seems to be included in the manuscript is “Supplementary 5. Figure 2” (one of them), appearing in line 174 as “Supplementary S4”. Line 112. “Supplementary S1” is lacking in supplementary files. Line 130. “Supplementary S2” is lacking in supplementary files. A complete and revised file of supplementary figures should be added before publication.

Dear Reviewer #2,

Thank you for considering that our manuscript has been adequately described.

We are sure that this study will improve.

1) To date there is an extensive bibliography regarding the influence of different light intensities in gene expression, growth rate and anthocyanin accumulation in several species of vaccinium (e.g. Zhang et al., 2022 (DOI 10.3389/fpls.2022.1073332); Guo et al., 2022 (https://doi.org/10.1186/s12870-022-03585-x); Wu et al., 2021(DOI: 10.1111/1755-0998.13467).

 Answer:  We agree with the suggestion, and as a result, the references and the approach regarding the role of the WRKY transcription factor in anthocyanin biosynthesis have been enhanced in the introduction section, incorporating findings from other studies.

2) Some of them have been mentioned in the manuscript in other paragraphs (Samkumar et al., 2021 (DOI: 10.1111/pce.14158)), but as the expression profiles of VmWRKY genes under different light conditions are being displayed in the manuscript, a deeper explanation of the importance of light should be included in the introduction.

Answer:  We agree with the suggestion, and as a result, the references and research focus have been revised to emphasize the involvement of the WRKY transcription factor in anthocyanin biosynthesis, as described in the introduction section.

3) Re-write the paragraph contained in lines 84-86 as it is confusing.

Answer: The paragraph has been rewritten.

4) 2.1 Identification of the WRKY protein family Line 112. I cannot find Supplementary S1 in the provided supplementary file.

Answer: The Supplementary 2 has been enhanced, expanded, and improved with better referencing.

5) 2.2 Analysis of cis elements in WRKY gene promoters A table (or graph) should be included providing relevant information such as frequency of cis-elements and a brief description of the biological pathways they are involved in (hormone responses, stress responses, metabolite biosynthesis, etc.) Line 130. I cannot find Supplementary S2 in the provided supplementary file.

Answer: Supplementary S2 provides information on the CIS elements.

6) 2.3 Phylogeny, gene structure and motif analysis of WRKY proteins Figure 1. - A brief description of the motifs could be included: § Identify conserved regions (WRKY elements, Leucine-zipper, zinc-finger, etc.) § Show an average sequence (or a frequency diagram) of other conserved regions that cannot be identified.

Answer: The required information is provided in Supplementary S3

7) Arabidopsis thaliana orthologs could also be included in the phylogenetic tree. It would help identify the closest orthologs of each gene and allow to go deeper in knowledge through bibliography if desired. You may see Baillo et al., 2020 (https://doi.org/10.1371/ journal.pone.0236651) as an example. - Line 210: UTRs are indicated by blue boxes (instead of green).

Answer: We agreed with the suggestion and made the necessary improvements. We generated phylogenetic trees with 72 protein sequences from Arabidopsis thaliana AtWRKY and 69 protein sequences from blueberry VmWRKY. All relevant information can now be found in Supplement 3, which also includes the study reference. Additionally, we covered changing the blue text in the boxes, which was previously green.

8) Table 2: Additional information such as the conserved heptapeptide class and the zinc-finger type of the protein could be included.

Answer: The required information has been considered and it   stay in the supplementary with complementary analysis.

9) 2.5 Number of transcripts and expression patterns of VmWRKY genes - Line 166, 167, 176, 228…: “the transcriptional expression of genes for each sample of abscisic acid-regulated anthocyanin biosynthesis…” is not an appropriate description of the samples in the study. The samples come from the experiments described in Samkumar et al., 2021 (DOI: 10.1111/pce.14158) where light quality produced differences in the expression of some ABA and anthocyanin biosynthetic genes. This description is confusing and it should be modified.

Answer: We acknowledge the observation and it has been improved, as a result we have revised the section regarding the involvement of the VmWRKY genes in light-induced biosynthesis.

10) This description is confusing and it should be modified. - A table identifying those genes that are common to every condition or unique should be added. - Line 174. “Supplementary S4”?? or Supplementary S5. Figure 2?? - Paragraph 175-183: It is confusing and should be re-written. - In my opinion the expression plots depicted in Figure 3 B and D do not offer extra information from what is represented in the heatmaps (Figure A and C).

Answer 1: We agree with the suggestion, the text was revised and clarified regarding: distribution of expression of the samples on a multi-dimensional scale -MDS, common expression of Vm WRKY genes in the samples, differential expression and highly expressed genes in all samples.

Answer 2: Complementary analyzes were also done that remain in supplements 5. However, figures 3B-3D representing highly expressed genes were removed and considered others (Fig.3).

11) Sample variability could be also visualized including individual samples, instead of the mean of expression between samples, in the heatmap. -Genes represented in Figure 3B do not seem to show “CLEAR differences in expression in conditions control, red and blue light” (line 229), and they are not relevant as light-sensitive genes at all.

Answer: The analysis has been clarified and the graphs have been better represented for better understanding in section 2.5 and Fig 3

12) In any case, what seems more important to me is the relative accumulation of each sequence compared to the control (differential expression) and not only if it is present or not in a sample. The ratio of expression should also be considered: genes with a significant differential expression between control and red/blue light conditions should be highlighted and identified in a separate table (e.g., VmWRKY42, VmWRKY58, VmWRKY43).

Answer: The analysis has been clarified and the graphs have been better represented for better understanding in section 2.5 and Fig 3

13) On the other hand, there is no correlation between the FPKM values in the bar diagrams in Figure 3D and the expression levels represented in the heatmap (Figure 3C). Probably it is because the sample order (left to right) is also different. Furthermore, the error bars in the plots seem to be wrong.

Answer: The analysis has been clarified and the graphs have been better represented for better understanding in section 2.5 and Fig 3

14) In my opinion the bar plots in Figure 3 could be removed. However, it could be interesting to show gene correlation between heatmaps instead, as one would expect that light-sensitive genes should show differential expression in different plant organs. For example, a co-expression network correlating lightsensitive genes with genes differentially expressed in plant organs; or a new clustering tree could be displayed integrating all 8 samples for light-sensitive WRKY genes.

Answer: The analysis has been clarified and the graphs have been better represented for better understanding in section 2.5 and Fig 3

15) 3.2 Analysis of cis elements in VmWRKY gene promoters. - Is there a correlation between WRKY groups and the nature/frequency of different cis-regulatory elements? It is particularly relevant for light responsive elements (see paragraph 270-273) and it could be informative as it could establish a relationship between light-mediated transcriptional regulation of WRKY genes and protein functional domains. It could be an interesting result to show.

Answer: A correlation analysis between the frequency of the cis regulators with respect to the VmWRKY genes was not carried out, however, within the supplementary 2, the frequency of presence of the elements was determined and a column was added, describing the group to which each VmWRKY gene belongs.

16) 3.3 Phylogeny, gene structure and motif analysis of WRKY proteins. - Paragraph 294-299: The numbers assigned to each group do no coincide with those showed in Figure 1. Furthermore, it seems to be redundant, as it has been previously explained in results. - Line 301. Please, use “abundant” instead of “expressive”.

Answer:  We agree with the suggestion, and the paragraph has been clarified.

17) 3.5 Phylogeny, gene structure and motif analysis of WRKY proteins. - Paragraph 355-357. It has no sense to me. Furthermore, the authors keep on using the wrong expression “abscisic acid-regulated anthocyanin biosynthesis”.

Answer:  We agree with the suggestion, and as a result, the references and research focus have been revised to emphasize the involvement of the WRKY transcription factor in anthocyanin biosynthesis, as described in the introduction section.

18) 4. Materials and Methods: 4.6 Transcriptomic analysis of VmWRKY genes - Line 425. Wrong reference: As far as I know, experiment PRJNA739815 is described in Wu et al., 2021(DOI: 10.1111/1755-0998.13467) - Line 430: Wrong reference: Experiment PRJNA747684 is described in Samkumar et al., 2021 (DOI: 10.1111/pce.14158) - Lines 432,433: Include the correct sample names for red and blue light experiments.

Answer: The suggestion was considered and the experimental design of red and blue light was complemented.

19) Supplementary files: Most of the supplementary figures provided with the manuscript (Supplementary 4. Figures 1, 2 and 3; Supplementary 5. Figures 2 and “also 2”) are not referenced. The only one that seems to be included in the manuscript is “Supplementary 5. Figure 2” (one of them), appearing in line 174 as “Supplementary S4”. Line 112. “Supplementary S1” is lacking in supplementary files. Line 130. “Supplementary S2” is lacking in supplementary files. A complete and revised file of supplementary figures should be added before publication.

Answer: All supplementary material and analysis contemplated, have been improved and carefully organized for better compression, files have been organized as one figure/table per file.  

Round 2

Reviewer 1 Report

I thank the authors for addressing my concerns and for explaining the reasons for their choices. I feel they adequately answered my questions.

just a few things, though: 

- please have a look at the last paragraph of the Methods, as it seems a leftover (and quite similar to the Conclusions)

Author Response

- please have a look at the last paragraph of the Methods, as it seems a leftover (and quite similar to the Conclusions)

Answer: Thank you for pointing our mistake. The paragraph was changed and moved to the end of the results.

Reviewer 2 Report

Dear authors,

Thank you for considering my recommendations, you have done a big effort and in my opinion the paper has been greatly improved. Congratulations.

However, I would like to mention some issues that are missing or could be modified in the manuscript

1) To date there is an extensive bibliography regarding the influence of different light intensities in gene expression, growth rate and anthocyanin accumulation in several species of vaccinium (e.g. Zhang et al., 2022 (DOI 10.3389/fpls.2022.1073332); Guo et al., 2022 (https://doi.org/10.1186/s12870-022-03585-x); Wu et al., 2021(DOI: 10.1111/1755-0998.13467).

 Answer:  We agree with the suggestion, and as a result, the references and the approach regarding the role of the WRKY transcription factor in anthocyanin biosynthesis have been enhanced in the introduction section, incorporating findings from other studies.

Answer 2: The introduction has been improved. However, you have not added none of the suggested papers, but it doesn’t mind, there’s no need to mention them.

4) 2.1 Identification of the WRKY protein family Line 112. I cannot find Supplementary S1 in the provided supplementary file.

Answer: The Supplementary 2 has been enhanced, expanded, and improved with better referencing.

Answer 2: Yes, it has been improved. But it is in the Supplementary 1. No matter, in the manuscript it is properly referenced… However, a reference to the actual Supplementary 2 table does not appear in the text.

5) 2.2 Analysis of cis elements in WRKY gene promoters A table (or graph) should be included providing relevant information such as frequency of cis-elements and a brief description of the biological pathways they are involved in (hormone responses, stress responses, metabolite biosynthesis, etc.) Line 130. I cannot find Supplementary S2 in the provided supplementary file.

Answer: Supplementary S2 provides information on the CIS elements.

Answer 2: You have done a pretty good job! However, the information regarding the frequency/quality of the cis-elements is provided in the Supplementary S3 to S6. Only S3 is referenced in the text, please, include the others.

7) Arabidopsis thaliana orthologs could also be included in the phylogenetic tree. It would help identify the closest orthologs of each gene and allow to go deeper in knowledge through bibliography if desired. You may see Baillo et al., 2020 (https://doi.org/10.1371/ journal.pone.0236651) as an example. - Line 210: UTRs are indicated by blue boxes (instead of green).

Answer: We agreed with the suggestion and made the necessary improvements. We generated phylogenetic trees with 72 protein sequences from Arabidopsis thaliana AtWRKY and 69 protein sequences from blueberry VmWRKY. All relevant information can now be found in Supplement 3, which also includes the study reference. Additionally, we covered changing the blue text in the boxes, which was previously green.

Answer 2: Thank you for including this figure in the manuscript. I think it will be helpful. It is represented in Supplementary 7 (not 3), but the manuscript presents the correct number.

18) 4. Materials and Methods: 4.6 Transcriptomic analysis of VmWRKY genes - Line 425. Wrong reference: As far as I know, experiment PRJNA739815 is described in Wu et al., 2021(DOI: 10.1111/1755-0998.13467) - Line 430: Wrong reference: Experiment PRJNA747684 is described in Samkumar et al., 2021 (DOI: 10.1111/pce.14158) - Lines 432,433: Include the correct sample names for red and blue light experiments.

Answer: The suggestion was considered and the experimental design of red and blue light was complemented.

Answer 2: The reference paper for Bioproject PRJNA747684 (number 87 in the manuscript) is not correct. On the other hand, the identification numbers of the 9 samples in the project are all identical (paragraph 4.6). Please, add the correct ones.

Author Response

Thanks for pointing that out. The few points remaining regarding the citation of suplementary files and other small mistakes were changed accordingly and left as yellow highlights in the text.